# Volatile Chemical Variation of Essential Oils and Their Correlation with Insects, Phenology, Ontogeny and Microclimate: *Piper mollicomum* Kunth, a Case of Study

**DOI:** 10.3390/plants11243535

**Published:** 2022-12-15

**Authors:** Daniel de Brito-Machado, Ygor Jessé Ramos, Anna Carina Antunes e Defaveri, George Azevedo de Queiroz, Elsie Franklin Guimarães, Davyson de Lima Moreira

**Affiliations:** 1Instituto de Biologia, Pós-Graduação em Biologia Vegetal, Universidade do Estado do Rio de Janeiro, Maracanã, Rio de Janeiro 20550-013, Brazil; 2Diretoria de Pesquisa do Instituto de Pesquisas Jardim Botânico do Rio de Janeiro, Jardim Botânico do Rio de Janeiro, Rio de Janeiro 22460-030, Brazil; 3Centro de Responsabilidade Socioambiental do Instituto de Pesquisas Jardim Botânico do Rio de Janeiro, Jardim Botânico do Rio de Janeiro, Rio de Janeiro 22460-030, Brazil; 4Instituto de Tecnologia em Fármacos, Fundação Oswaldo Cruz, Manguinhos, Rio de Janeiro 21041-250, Brazil

**Keywords:** chemical ecology, essential oils, terpenes, pollination, ecological interactions, Piperaceae

## Abstract

The aim of this study was to monitor the volatile chemical composition from leaves and reproductive organs of *Piper mollicomum* Kunth (PM), in its reproduction period, as well as register inflorescence visitors, microclimate and phenological information. The essential oils (EOs) obtained from the different fresh organs by hydrodistillation were identified and quantified by Gas Chromatography/Mass Spectrometry (GC/MS) and by GC coupled to a Flame Ionization Detector (GC/FID), respectively. The cercentage content of some volatiles present in reproductive organs, such as limonene, 1,8-cineole, linalool and eupatoriochromene, increased during the maturation period of the inflorescences, and decreased during the fruiting period, suggesting a defense/attraction activities. Furtermore, a biosynthetic dichotomy between 1,8-cineole (leaves) and linalool (reproductive organs) was recorded. A high frequency of bee visits was registered weekly, and some correlations showed a positive relationship between this variable and terpenes. Microclimate has an impact on this species’ phenological cycles and insect visiting behavior. All correlations between volatiles, insects, phenology and microclimate allowed us to present important data about the complex information network in PM. These results are extremely relevant for the understanding of the mechanisms of chemical–ecological plant–insect interactions in Piperaceae, a basal angiosperm.

## 1. Introduction

*Piper mollicomum* Kunth (Piperaceae) is a heliophilous or cyophilous species, which means it can grow both in high luminosity environments and in humid and shady places. It has membranous, tomentose leaves and spicule inflorescences [1,2]. This plant blooms in many locations across Brazil typically from September to January [1,2,3]. This wide distribution leads to different popular names, such as aperta-ruão, jaborandi, jaborandi-manso, pariparoba and rabo-de-galo. Because of its pantropical range and lack of preference for any particular temperature zone, this species has been documented in a variety of vegetation types and climatic conditions, including antropic area, riparian forest, semi-deciduous seasonal sorest and ombrophilous forest [1,3,4,5].

The bio-pharmacological effects of *P. mollicomum* are widely recognized, and include increased gastrointestinal motility [6], antifungal [7,8], antibacterial [9], larvicidal [10] and antinociceptive activities [11]. Some ethnobotanical studies also suggest that different organs of this plant are used to treat back pain, to decrease menstrual flow, and even as a hair tonic [12,13,14,15].

In addition to its medicinal applications, specimens of *P. mollicomum* are used as mystical resources to achieve spiritual contact with deities in rituals of African-based religions. Amaci is one of these important rites [16].

This species also contributes to several biotic interactions. For example, its infructescences have already been described as the main food resource for two species of frugivores bats, *Carollia perspicillata* (Linnaeus., 1758) and *C. castanea* (Frederirick, 1890) [17,18]. Infructescences are critical for the reproductive success of these animals and of several species of frugivorous birds [19,20]. Because infructescences are available all year long, *Piper* L. specimens may efficiently disseminate their seeds in their habitat with the aid of these animals. [21,22]. In this sense, current studies have focused on understanding chemical communications between *Piper* and animals [23].

Some authors point out that the reproductive organs are anatomical structures characterized especially as mutualistic in their ecological relationships [2,24]. However, other conceptions propose a model in which these structures act as hotspots in mutualistic and antagonistic interactions, mainly due to their great phytochemical diversity [23,25,26,27,28,29,30,31,32,33]. According to several studies, EOs from fruits and seeds have a variety of specialized metabolites (SMs) that differ from those found in other vegetative tissues [23,34,35,36,37]. Recent studies have shown that amides and 7-desmethylenekecalin have high toxicity against opportunistic pathogens in immature fruits [3,36,38,39,40]. On the other hand, amide concentrations decreased in the pericarp of ripe fruits, making these organs more attractive to seed dispersers. This phenomenon shows how biotic influences can affect the composition of SMs found in various plant tissues [2,23,37,41].

Despite the great entomological biodiversity in neotropical forests, many plant specimens are preyed by specific groups of insects [42,43]. This homogeneity is mainly generated by biotic filters that culminate in a specific herbivory between phytophages and plants [24,44,45]. Some of these trophic interactions are observed in *Piper* species and *Eois caterpillars* (Hübner., 1818) [24,42,45,46,47,48,49,50,51,52]. For example, leaves of *P. mollicomum*, that are continuously defoliated by herbivores, emit EOs attractive to predatory wasps who feed on their herbivorous rivals, promoting negative pressure on them. These damaged organs released terpenes that were detected in very low percentages in intact leaves. This mechanism infers that some constituents may benefit the plant by attracting opportunistic insects, such as parasitic predator wasps, which attack their natural enemies [24].

Essential oils are complex mixtures of volatile and lipophilic substances, biosynthesized in various plant organs within intracellular compartments. These volatiles are composed of a wide variety of complex compounds, however, the main constituents in these mixtures are terpenes, organic substances derived from isopentyl diphosphate (IPP) or 3,3-dimethylallyl diphosphate (DPP) units [53,54,55,56]. These metabolites are composed of five carbon atoms and are subsequently biotransformed by enzymatic activity [3,53,54,55,56]. The most common terpenes in Piperaceae are monoterpenes and sesquiterpenes, although there are records of volatile diterpenes [57]. Two important biosynthetic pathways lead to the production of monoterpenes and sesquiterpenes: acetate-mevalonate and methylerythritol-phosphate [3,53,54,55,56].

Several studies correlate the participation of substances that can be found in EOs in some important ecological relationships [2,53,55,58,59,60,61,62,63,64,65,66]. For example, some volatiles have a high repulsive toxic action, causing food aversion to herbivores of the plants that produce them [51,52,67]. In addition, volatiles can even prevent the oviposition of insects and mites, and attract their natural enemies [52,68,69].

Not only disharmonious relationships predominate between *Piper* species and in-sects. It is important to note the harmonious relationships between these two groups, particularly pollination, a subject that has received regular attention in recent years [70,71,72,73].

Ramos et al. (2021) analyzed the chemical phenotypic plasticity of EOs from *Piper gaudichaudianum* Kunth at different seasons and periods of the year. This study made it possible to propose a new chronotype, in addition to other important findings, such as a new micromolecular index that allows inferences about the reduction–oxidation of mixtures (Ramos and Moreira Index) [55].

Furthermore, some studies were carried out by our group with *P. mollicomum* that showed a high percentage of linalool, eupatoriochromene and *E*-nerolidol in the reproductive structures [2,3]. However, in these studies, macroclimatic data were used, and the different phenophases of the reproductive organs were not evaluated [2]. The correlations between volatiles, insect visitation, microclimate and reproductive organ ontogeny were studied [74,75,76,77,78,79,80], therefore, a much more embracing work, and of great chemical-ecological importance. The results gathered are of essential importance for understanding the processes of chemical–ecological interactions in Piperaceae since this is the first scientific study to completely examine several factors that interact with this species of *Piper* throughout its reproductive period.

## 2. Results

The EOs of the stages of the reproductive organ showed variation in their percentage of volatile chemical compositions (0.01% to 1.47%; *v*/*w*). Stages 1, 2, 3 and 4 presented the lowest percentages (0.01% to 1.12%; *v*/*w*), and stage 5, the highest (1.2% to 1.47%; *v*/*w*) (Appendix A). The volatile constituents were different in most of the investigated months, showing a rich and diversified fraction in non-oxygenated sesquiterpenes in the leaves and oxygenated monoterpenes in the five stages of the reproductive organ.

In the initial months of flowering (September and October), the oxygenated monoterpene linalool was the chemical constituent that registered the highest relative percentages, both in leaves and in most stages of the reproductive organ (September—29.23% to 73.13%; October—27.20% to 51.96%). Also noteworthy is the non-oxygenated monoterpene limonene, which showed high percentages in most EOs in this period, including in the reproductive organs (0.01% to 20.07%). In November, the major constituent in the leaves was identified as the oxygenated monoterpene *α*-terpineol (11.41%), and the constituent with the most prominent relative percentage in the different stages of the reproductive organ was 1,8-cineole (26.07% at 58.77%). In December, the constituents that stood out in the foliar EOs were the non-oxygenated monoterpenes *α*-pinene (11.30%) and *β*-pinene (6.49%), the oxygenated monoterpene linalool (11.28%) as well as the non-oxygenated sesquiterpene *β*-elemene (8.62%). On the other hand, in the different stages of the reproductive organ, the main identified constituent was 1,8-cineole (22.96% to 32.01%), in addition to *α*-pinene (3.50% to 14.7%) and linalool (0.00% to 10.46%). In the last month of this research (January) 1,8-cineole was the constituent that presented the highest relative percentages in the EOs, not only in the leaves (8.62%), but also in the different stages of the reproductive organ (13.50% at 49.32). It is important to highlight that the constituent desmethylencecalin (eupatoriochromene) showed a significant difference (*p* < 0.05) in the relative percentage of leaves (0.00% to 2.56%) and stages of reproductive organs (0.12% to 32.92%) from September to January.

In addition to presenting the major constituents of the last stages of floral development, Appendix A shows Spearman’s correlation (*r*^2^) between EOs’ constituents vs. seasonality. For this analysis, we highlighted all the constituents of the EOs that were identified at least once in the periods and stages of the analyzed reproductive organs. The results show negative correlations for the constituents eupatoriochromene (*r*^2^ = −0.755, *p* < 0.05), *α*-pinene (*r*^2^ = −0.878, *p* < 0.05), and myrcene (*r*^2^ = −0.799, *p* < 0.05). This information infers that there are changes in the content of EOs from the first stage of floral development to the beginning of fruiting, a fact that leads to a significant decrease in these compounds, while the development of reproductive structures occurs. As result of this Spearman’s correlation, we proposed to investigate the trend on the ontogenic pattern of the percentage contents of the main constituents of the EOs along each stage of the reproductive organ (Figure 1). For this correlation, we selected only the constituents that registered percentage contents above 5% at least once in the EOs in the periods and stages of the analyzed reproductive organs. The results showed that the relative percentages of *E*-nerolidol, 1,8-cineole and linalool undergo increases from stage 1, 2 (period of maturation) and 3 (anthesis), and decreases in the period of fructification (stages 4 and 5). The graphs (Figure 1) denote that the compounds limonene and eupatoriochromene present consistency in their percentage contents in the first stages of floral maturation and decrease in the stages of fructification. Besides that, *α*-pinene was the only one that showed a decrease in its relative percentage content from the first to the final stages of floral development.

Figure 1 illustrates the main constituents of the EOs of the reproductive structures that increase in the flowering stages and decrease in the fruiting stages, exception for α-pinene. Henceforth, we did principal component analyses (PCA) to assist in the interpretation of the distinctions of the chemical constitution (Figure 2). The variables’ projection factors underwent a total change of 86.06%, with the contribution of the first principal component (PC1) of 47.38%, and 38.68% for the second principal component (PC2). Two groups promoted significant variations in the content of their components: (1) predominant in linalool and (2) predominant in 1,8-cineole. The findings demonstrate that leaves and reproductive organs were separated at the start of blooming in September and October due to an increase in PC2’s positive charge (2), influenced by the high percentages of linalool (PC2, 38.68%). Due to a rise in the percentage content of 1,8-cineole (PC2, 47.38%), leaves and reproductive organs displayed an increase in the positive charge of PC1-(1) from the peak of blooming till fructification between November and January. At the peak of flowering until fructification (November to January), leaves and reproductive organs showed an increase in the positive charge of PC1-(1), due to the increase in the percentage content of 1,8-cineole (PC2, 47.38%).

Appendix A also demonstrates the similarity of the chemical profile biosynthesized in the EOs (leaves and reproductive organ). The results show that at the beginning of flowering, the characteristics of the volatile constituents presented greater chemical similarities, both for leaves and for reproductive organs. However, from the floral apex to the final stages of fruiting, the chemical profile was different, demonstrating that the compounds are more distinct (Appendix A). Finally, Appendix A presents the Euclidean Hierarchical Grouping, which clearly demonstrates the biosynthetic dichotomy of the volatile constituents from leaves and reproductive structures analyzed at the beginning (linalool-rich samples) and at the end of flowering (1,8-cineole-rich samples).

In order to evaluate correlations between the chemical constituents of the EOs and the frequency of visits by potential pollinators, weekly observations were carried out. Data collection indicates that bees and flies were the main groups of insects observed in all inspections (Appendix A). In total, the insects visited the mature inflorescences 4332 times in the analyzed period. Bees were the most noticed, with high frequencies of visits almost every day. Insect 2 (Hymenoptera—*Tetragonisca angustula* Latreille, 1811) was the floral visitor that registered the highest frequency (*n* = 3042). Insect 6 (Hymenoptera—*Colletidae* sp.), another bee, was the second hymenopteran with the highest number of visits (*n* = 820). The other bees, insects 5 (Hymenoptera—*Halictidae* sp. 2) and 4 (Hymenoptera—*Halictidae* sp. 1), recorded frequencies of 238 and 67, respectively. Diptera had significantly different visit rates among them. While insect 3 (Diptera—*Syrphidae* sp. 2) was the third insect with the highest number of visits (*n* = 164), insect 1 (Diptera—*Syrphidae* sp. 1) was observed only once (*n* = 1). The data of the weekly sums referring to all frequencies of visits are described in Appendix A.

Canonical correspondence analysis was carried out to assess a potential correlation between insect visits and volatile components (Figure 3 and Figure 4). This analysis is more accurate than Spearman’s correlation, as it links all variables, including visitation rates among insects. The first graph (Figure 3) indicates correlations between leaf constituents’ data vs. insect visits to mature inflorescences. The results demonstrate a strong positive correlation between the frequency of visits and the percentage content of non-oxygenated monoterpenes camphene, limonene and *E-β*-ocimene; in addition to the acyclic oxygenated monoterpene linalool. The results also indicate a strong positive correlation between sesquiterpenes *E*-caryophyllene and aromadendrene, with the visits of the insects to inflorescences. The graph also shows that the constituents that can attract insects were biosynthesized mainly in the first months of observation, September and October, when the first stages of inflorescence sprouting began. Concurrently, the sesquiterpenes *E*-nerolidol and *β*-selinene, in addition to the monoterpenes *E-β*-ocimene, *Z-β*-pinene, *α*-pinene, 1,8-cineole, as well as eupatoriochromene, did not show a significant positive correlation with respect to the constancy of visits. Additionally, Figure 3 demonstrates that during the inflorescence and infructescence maturation periods (November to January), the components that had a negative correlation with the frequency of insects were biosynthesized. Figure 4 indicates correlations between data from constituents of mature inflorescences vs. insect visits. These analyses present a strong positive correlation between the frequency of visits and percentage content of non-oxygenated monoterpenes such as camphene and *Z-β*-ocimene, in addition to the acyclic oxygenated monoterpene linalool, and the sesquiterpenes *E*-caryophyllene and aromadendrene. The constituents *E*-nerolidol, *β*-selinene, *α*-pinene, 1,8-cineole, *E-β*-ocimene, limonene and eupatoriochromene did not show significant correlations with the variables in the analysis.

These results show that some major constituents present in the EOs may influence the frequency of visits by potential pollinators to stage 3 (anthesis) of *P. mollicomum*, as confirmed in the two canonical correspondence analyses. These data denote the value of terpenes in this important ecological function. Furthermore, these patterns enable us to deduce the possibility that the compounds that increase insects’ attraction to *P. mollicomum* inflorescences may have a synergistic effect. It is known that the volatile constituents of leaves, and not only those of inflorescences or mature flowers, influence the attraction of insects [24]. These compounds interact in a blend or volatile mixture, a phenomenon that was already described. However, we define it as volatile fog (Figure 5).

In addition to canonical correspondence analyses on the insect–plant interaction in *P. mollicomum*, we performed Spearman’s correlation analysis to confirm the relationships between visitors and volatile constituents vs. abiotic microclimate factors (Appendix A). The analyses show strong positive correlations between the following constituents in the attraction of insects: camphene, eupatoriochromene, *α*-pinene, aromadendrene and camphor. The data also show a moderate positive correlation between the ambient temperature (°C) (*r*^2^ = 0.749, *p* < 0.05; *r*^2^ = 0.633, *p* < 0.05; *r*^2^ = 0.606, *p* < 0.05; *r*^2^ = 0.642, *p* < 0.05), inflorescences temperature (*r*^2^ = 0.688, *p* < 0.05; *r*^2^ = 0.559, *p* < 0.05; *r*^2^ = 0.606, *p* < 0.05; *r*^2^ = 0.616, *p* < 0.05) and leaf temperature (*r*^2^ = 0.763, *p* < 0.05; *r*^2^ = 0.626, *p* < 0.05; *r*^2^ = 0. 717, *p* < 0.05; *r*^2^ = 0.692, *p* < 0.05) for insects 2 (*Tetragonisca angustula,* Latreille. 1811), 3 (*Syrphidae* sp. 2), 5 (*Halictidae* sp. 2) and 6 (*Colletidae* sp.), respectively. A weak positive correlation was jointly demonstrated for light intensity and frequency of visits by insects 4 (*Halictidae* sp. 1) and 5 (*Halictidae* sp. 2); (*r*^2^ = 0.538, *p* < 0.05; *r*^2^ = 0.560, *p* < 0.05). This information allows us to infer that insects are more attracted to plants in hot periods. This fact is probably due to the intense volatilization of the constituents that possibly attract insects.

In contrast, our findings also describe a moderate negative correlation between relative air humidity (*r*^2^ = −0.667, *p* < 0.05; *r*^2^ = −0.622, *p* < 0.05) and rainfall index (*r*^2^ = −0.649, *p* < 0.05; *r*^2^ = −0.657, *p* < 0.05; *r*^2^ = −0.649, *p* < 0.05) with the frequency of insect visits 2 (*Tetragonisca angustula,* Latreille. 1811), 3 (*Syrphidae* sp. 2) and 6 (*Colletidae* sp.), respectively. These findings support earlier research that indicated that increase in humidity impairs bee visits to flowers of different plant types.

It is known that the meteorological microclimate variables may influence the attraction of pollinators, however, in addition to this important information, our purpose was to assess whether these factors also influence the biosynthesis of chemical constituents identified in the EOs of *P. mollicomum*. Appendix A shows Spearman’s correlations between the measured microclimate vs. percentage content of the major compounds in the reproductive organs of *P. mollicomum*. The mature inflorescence (stage 3—anthesis, Figure 6) was the structure chosen for the analysis of these correlations because it was the only one of the five stages visited by all the insects recorded in this research. The results demonstrate that camphene (*r*^2^ = 1.000, *p* < 0.14), aromadendrene (*r*^2^ = 1.000, *p* < 0.31), camphor (*r*^2^ = 1.000, *p* < 0.21), *α*-terpinene (*r*^2^ = 0.935, *p* < 0.21), tolualdehyde (*r*^2^ = 0.935, *p* < 0.21), and *δ*-cadinene (*r*^2^ = 0.926, *p* < 0.24) showed positive correlations with the environmental variable wind speed (m/s). The data also describe that both the temperature of the inflorescences in stage 3 and the temperature of the leaves (°C) can negatively influence the percentage of *β*-pinene (*r^2^* = −0.945, *p* < 0.02; *r^2^* = −0.916, *p* < 0.02, respectively). According to our results, rain and relative humidity have a strong influence, directly proportional, on the chemical constitution, mainly *α*-pinene (*r*^2^ = −0.880, *p* < 0.23), *β*-selinene (*r*^2^ = 0.985, *p* < 0.23), myrcene (*r*^2^ = −0.927, *p* < 0.26), *δ*-cadinene (*r*^2^ = 0.926, *p* < 0.23), tolualdehyde (*r*^2^ = 0.935, *p* < 0.23), *γ*-terpinene (*r*^2^ = 0.935, *p* < 0.23), benzyl benzoate (*r*^2^ = 0.935, *p* < 0.23) and eupatoriochromene (*r*^2^ = 0.911, *p* < 0.23).

As mentioned, we verified that the volatile constituents could instill the frequency of insect visits, probably by olfactory attraction. In order to enrich our results, we propose to investigate whether this activity of potential pollinators is influenced by visual attraction. For this, we performed Spearman’s correlation analysis on the variables of insect visits vs. pattern of phenological events in *P. mollicomum* (Appendix A). Surprisingly, the analysis did not show significant results for these correlations.

We also performed Spearman’s correlations between abiotic factors of microclimate vs. pattern of phenological events (Appendix A). Data showed a strong negative correlation between leaf fall and temperature (°C) (*r*^2^ = −0.783, *p* < 0.05). Additionally, there is a strong positive correlation between rainfall (mm) and leaf sprouting (*r*^2^ = 0.800, *p* < 0.05). This correlation indicates that the greater the rainfall, the more pubescent leaves initiate the sprouting process. Moreover, data showed that the leaf sprouting phenophase is negatively correlated with the microclimate light intensity (*r*^2^ = −0.800, *p* < 0.05). This trend indicates that higher luminosity inhibits the sprouting process. However, this correlation needs to be further evaluated in detail, as the opposite is expected [81]. We speculate that there could be an optimal luminosity for budding in *P. mollicomum*, which would support these results. These issues can be addressed by experiments using carefully regulated in vitro culture.

## 3. Discussion

During the observation period of this research, we examined and described five stages of ontogenic development of the reproductive organ of *P. mollicomum*. Four of the five were registered between September and November 2020. The last stage was only found in the last two months of the analysis (December 2020 and January 2021). The results show that at the beginning of flowering, there was mainly biosynthesis of the volatile constituents linalool and limonene. In this period, the plants still had their immature inflorescences, with the absence of enough pollen grains to favor the pollination process (and consequent reproduction). With that, the fundamental issue to keep the specimens viable for propagation is the defense of these relevant and important organs. Some studies show that plant species exude toxic volatiles, against herbivores, most markedly in the early stages of flowering, mainly to maintain the accessibility of their buds and, as a result, ensure successful reproduction [82,83]. Zheng et al. (2020) in a study carried out with specimens of lavender (*Lavandula angustifolia* Mill., Lamiaceae), tobacco (*Nicotiana tabacum* L., Solanaceae) and mint (*Mentha spicata* L., Lamiaceae), showed that genes that significantly contribute to protection from different reproductive organs were mainly stimulated during the early stages of pubescent tissue development. Another study related to the defensive behavior of Japanese pepper (*Zanthoxylum pipertum* L.D.C., Rutaceae) corroborates these findings [84].

The initial stages of development of reproductive organs tend to be stressful for many plant species, as in these stages, plants need to reallocate their energy reserves, both for the growth of their flowers or inflorescences, and for the development of defensive strategies. There is an intense investment of plant metabolism in the biosynthesis of compounds of primary and secondary metabolism [82]. The study by Zheng et al. (2020) also reported that the constituent limonene, biosynthesized in tobacco leaves, acted as a repellent against aphids (*Myzus persicae* Sulzer., 1776). This article explains that limonene also protects plants against herbivory by attracting ladybirds (*Harmonia axyridis* Pallas., 1773), natural enemies of aphids [83]. These references corroborate our findings and suggest the hypothesis that limonene may act in a plant–herbivore–carnivore tritrophic interaction [83,85]. These findings, regarding the induction of chemical signals provided by limonene, were also demonstrated in studies with Fabaceae, such as soybean (*Glycine max* (L.) Merrill). Coleoptera of the species *Coccinella septempunctata* Linnaeus., 1758, were attracted to plants that contained considerable amounts of limonene and that were extensively damaged by aphids (*Aphis glycines* Matsumurae., 1917) [86]. In addition to beetles, some findings explain that limonene can attract predatory aphids, such as *Phytoseiulus persimilis* Athias, 1957, and *Neoseiulus californicus* McGregor, 1954, protecting specimens of Lamiaceae (*Mentha spicata* L.) that emit volatiles against herbivory caused by insects [52]. Other studies showed that a simultaneous effect between the constituents carveol, pinene and limonene act as promising repellents [87,88]. However, some studies carried out with limonene have reported the opposite effect regarding its repellent activity against herbivorous insects. Limonene can attract pathogens to the leaves of orange trees (*Citrus sinensis* L., Rutaceae) and, consequently, induce an attack by herbivores that directs a negative trait to these plants [89]. However, this latest finding does not invalidate the repellent effect of limonene. Some species may adapt against the adversities in their niche and escape the repellency or tritrophic activity evidenced by the biosynthesis and emission of limonene [83]. For example, one study showed that *Musca domestica* L. can reduce limonene toxicity by catalytic activity, converting this monoterpene into less toxic compounds [90].

As mentioned, another biosynthesized compound in the early stages of flowering of *P. mollicomum* was the monoterpene linalool. Previous studies have already suggested that this constituent would be able to help attract insects [91]. According to the authors, a relevant percentage content of this monoterpene was found in the mandibles of Hymenoptera (genus *Colletes*). Approximately 30 years later, some scientists reinforced this hypothesis by demonstrating that, in addition to linalool, the cyclic monoterpene 1,8-cineole, could also aid in the pollination process by attracting bees [92]. These data agree with our findings, which recorded high percentage amounts of these monoterpenes in reproductive organs; together with high frequencies of Hymenoptera visits. Six years later, it was shown that specimens of *Lippia alba* Mill. Brown., when exposed to high temperatures, volatilize a higher content of linalool [62]. Glinwood and Blande (2016) subsequently demonstrated that linalool and 1,8-cineole were identified in high percentages in the leaves of plant species when subjected to abiotic stressors [93]. Recently, chemical–ecological analyses carried out by our group, revealed a strong positive correlation between bee visits to mature inflorescences and the percentage of linalool in *P. mollicomum* specimens [2]. Linalool is common in EOs of reproductive organs of about 70% of angiosperms [94], explaining that this monoterpene may play a key role in the metabolism and ecological relationships of these plants [2,62,91,92,93,95,96,97].

It has been known for years that the enzymes of the linalool–synthase complex facilitate the production of linalool throughout the ontogeny of flowers [96,98,99,100,101]. In addition to the attractive functionality related to pollinating insects, linalool may have an effective defensive effect against herbivores [102,103,104]. A recent study with a plant native to Europe, *Arabidopsis thaliana* (L.) Heynh (Brassicaceae), showed promising evidence. Besides, recent findings showed that linalool may not have a direct role in plant defense, but an indirect importance in this applicability by the fragmentation of metabolites derived from its oxidation. The work by Boachon et al. (2015) described that one of the enzymes that plays a fundamental role in the metabolism of linalool, CYP76C1, may be involved in its oxidation process in *A. thaliana* flowers. This CYP catalyzes a metabolic cascade for the formation of soluble oxidizing derivatives, such as 8-carboxy-linalool, in addition to volatile cyclic derivatives, such as alcohols and aldehydes [85]. This study also showed other interesting data about the CYP76C1 enzyme complex. Surprisingly, this CYP76C1 is located almost exclusively in the stamens (anther and filament) and flower petals. The most curious is the fact that CYP76C1 is prominently expressed in floral tissues when floral anthesis occurs, that is, in the early stages of flowering, a period in which reproductive structures are more visited by both harmonic and disharmonious insects [85]. In the first flowering periods, there is a decrease in linalool content due to its oxidation, which is related to the higher expression of CYP76C1 in floral structures [85]. These authors also tested whether linalool derivatives in *A. thaliana* flowers altered the behavior of adverse visitors. The results of these tests showed that thysanopterans, in addition to other herbivorous insects, have a greater predilection for plants with a high content of linalool than for those with derivatives of this monoterpene, demonstrating the repulsive role of linalool oxidation products [85]. It is already known that enzymes of the CYP76 subfamily have been recorded in other plant species, acting on different substrates, and promoting their multifunctionality; however, always causing the biosynthesis of antipathogenic compounds [105,106,107]. Thus, linalool may be involved in two ecological purposes: attracting pollinators and protecting plants from pests, which is a well-known illustration of how nature seizes opportunities. The biosynthesis of compounds requires a high expenditure of energy. The oxidation of linalool orchestrated by enzymes of the CYP76 subfamily showed the malleability of plant metabolism in responding to its niche. These findings corroborate our results, which demonstrate a decrease in the percentage of linalool, from the initial stages of flowering to the final stages of inflorescence development (fruiting). This assumption could be crucial for creating new organic pesticides that will help decrease the number of hazardous insecticides and pesticides, that are now widely used in agriculture. This is a typical illustration of how observing ecological interactions may assist in the development of new biomolecules.

Mature inflorescences were noticeably more prevalent from November 2020 to January 2021 so that the biosynthesis of components that aid in luring prospective pollinators would be more feasible and support the reproduction process [2]. However, as mentioned earlier, some constituents may induce the attraction of insects, as well as herbivores and pests [85,89,102,103,104,105,106,107]. At this point, plants faced some difficulties: should they generate compounds that cause harmonic attractions or that defend vegetative structures against natural enemies?

The important data presented in the bar graphs (Figure 1) show the trend of the ontogenic pattern of the percentage contents of the main constituents of the EOs: some constituents undergo increases in their percentages during the period of inflorescence maturation (September and October) and decreases during the fruiting period (November to January). In other words, these constituents may be related to the defense process of reproductive structures, as there are decreasing changes in their contents, from the first stage to the end [51,83,85,108]. At this point, the increase and consequent decrease in the relative percentage of eupatoriochromene stands out. Some chromenes can act in the defense of vegetative organs against natural predators or phytopathogens [8,109,110,111,112]. In addition, eupatoriochromene may be related to the process of maturation and development of inflorescences. This hypothesis is supported by published articles on the effect of eupatoriochromene on ontogenesis [113], as well as other chromenes, including in Piperaceae species [114,115]. Still, in the late 1980s, a study suggested that chromenes can act both in promoting the development of the reproductive structures of plant species, and jointly, in promoting the protection of floral organs [116,117]. In that work, which analyzed the metabolism of *Ageratin adenophora* (Spreng.) R.M.King and H.Rob (Asteraceae) throughout its life cycle, high contents of hydroxylated derivatives of desmethoxyencalin (eupatoriumchromene and enkekalin) were shown to influence the ontogenicity of floral organs in the early stages [116]. Another study that analyzed the ontogenic development and chemical constitution of EOs present in *A. adenophora* leaves identified high percentages of chromene derivatives in pubescent leaves and low percentages in senescent leaves [117]. The relevance of chromenes in the role of ontogenic development is not only associated with flowers. In a recent study, the growth of vegetative organs, such as roots and calluses, was inferred for enkekalin [118]. These data corroborate our study, since the EOs of senescent leaves showed percentage levels of eupatoriochromene not detected, or in insignificant values. The articles listed above also corroborate our analysis of the predominance of sesquiterpenes such as *E*-nerolidol, *β*-elemene and *D* germacrene in senescent leaves of *P. mollicomum*, which allows us to infer about the protective role of these constituents in leaf structures. As the eupatoriochromene content is generally higher in immature leaves, this compound may assist both in the development process and in leaf protection [117,118,119]. Our data show that higher levels of sesquiterpenes are present where lower percentages of eupatoriochromene are recorded. This biosynthetic dichotomy may be explained by the role of the foliar ontogeny of eupatoriochromene, and in the protection against herbivory by sesquiterpenes. Further research is necessary to determine if *P. mollicum* leaves exhibit this intriguing dichotomy in the time (circadian cycle and seasonality) and space (various compartments) of chromene/sesquiterpene biosynthetic activity. Besides, eupatoriochromene had already been identified as the main constituent present in the EOs of dry leaves and inflorescences of other *Piper* species [120,121,122].

Returning to the discussion started earlier and, as a reinforcement of the idea, our dataset makes us reflect on the presence, besides eupatoriochromene, of other constituents present in EOs, such as *E*-nerolidol, 1,8-cineole, linalool, limonene and α-pinene. As their respective percentages in the EOs of the reproductive parts decreased proportionally to the ontogenic development of the inflorescences, our inference is that these compounds act in a synergistic effect to assist in the process of protecting these organs against herbivores, which may cause, for example, damage to the palatability of these structures. It is known that fruits and infructescences can be preyed on by animals, which leads to seed dispersal. Toxic compounds present in these reproductive organs are important in the early stages of development, as they help to protect them [108,123]. However, this function loses its purpose in the mature stages [108,123]. It is possible that this causes a decrease in the content of these compounds. Therefore, two forces may act for the balance in space and time of production/accumulation of eupatoriochromene, *E*-nerolidol, 1,8-cineole, linalool, limonene and *α*-pinene in *P. mollicomum*: development and protection.

It must also be considered that infructescences and fruits accumulate a greater content of primary metabolites, therefore, a drainage effect must also be considered [124]. Some studies infer about a possible positive correlation between the chemical constituents present in infructescence’s EOs and the frequency of visitation by seed dispersers [123]. Among the many visitors to these reproductive parts, bats and birds play a prominent role in contributing to the propagation and germination of different plant specimens in natural ecosystems [125]. Recent studies have demonstrated the relevance of this ecological interaction between dispersers and plants, especially considering the analyses of management techniques and conservation strategies in forests that have suffered anthropic impacts [108]. It is known that 1,8-cineole can act as a general toxic substance in several types of herbivores or frugivores, such as skunks, brown galago (*Otolemur crassicaudatus* Geoffroy, 1812) and kangaroo species, such as *Macropus rufogriseus* Desmarest, 1817, causing damage to the digestive process of these marsupials [51,126,127]. It was inferred that 1,8-cineole can cause antimicrobial effects that promote changes in the microbiota of mammals. This terpene promotes changes in the fermentation process of the intestine of these herbivores, impairing digestibility and even interfering with the absorption of ingested nutrients [51,128,129,130]. Some herbivores, such as the koala (*Phascolarctos cinereus* Goldfuss, 1817), manage to reverse the toxicity of 1,8-cineole, and ingest a significant content of leaves that contain a high content of this monoterpene in their vegetative structures, such as species of the genus *Eucalyptus* L. [75,131]. According to research on the intestinal enzymes of koalas, each protein involved in the metabolic pathways of 1,8-cineole oxidation in these marsupials’ digestive systems operates in a very particular manner, reducing the toxicity of this monoterpene’s byproducts [131]. However, this constituent did not show a correlation with the frequency of insect visits despite being a predominant constituent in the EOs of different structures and stages of the reproductive organs from *P. mollicomum*. Our hypothesis is that 1,8-cineole is related to the repellency process of herbivores because there is a gradual increase in the early stages of inflorescence development and a significant decrease in the fruiting stage.

The results of this manuscript demonstrated by canonical correspondence and Spearman’s correlation, that linalool may play a major role in attracting potential pollinators, while 1,8-cineole is important in herbivore repellency. This evidence helps to support the hypothesis that in the reproductive period there is a pattern of conversion of chemical resources in the different organs, that is, 1,8-cineole in the leaves and linalool in the reproductive organs [3]. This is a typical case of compartmentalization with a view to protection/reproduction. In other words, during the reproductive stage, chemical components function to both attract pollinators and protect plant organs against herbivory. [2,3,24,51,62,74,75,79]. *Therefore, from the foregoing, it appears that the constituents in the EOs of P. mollicomum may present activity in attracting potential pollinating insects (linalool); attracting natural enemies of herbivores (limonene); participating in the ontogenic development of inflorescences, and possibly of leaves (eupatoriochromene); and acting in the defense of vegetative structures (eupatoriochromene and 1,8-cineole).*

However, our research was also designed to understand another question: is the biosynthesis of these constituents influenced by abiotic microclimate factors? Our results related to the Spearman’s analysis showed a negative correlation between inflorescence temperature vs. percentage content of cyclic monoterpene *β*-pinene. Previous studies mention the action of temperature and radiation on the *β*-pinene content [132]. Our results are also in agreement with previous studies that showed the effects of radiation and temperature on the degradation of *β*-pinene isomers [133,134,135]. These studies exposed specific definitions of temperature and relative humidity variations on possible changes in *α*-pinene content and found that this monoterpene undergoes EOs significant deterioration by photooxidation when exposed to high temperatures [134,135]. Considering light radiation, it is known that enzymes to produce terpenes are influenced by light such as 1,8-cineole synthase and linalool synthase. The analyses of volatile constituents in three stages of development in vitro, and in the field, of *Vitis vinifera* L. (Vitaceae), showed that when there was an increase in UVB radiation, there was also a significant increase in the percentage of oxygenated cyclic monoterpenes such as 1,8-cineole [136]. In another investigation, on the action of light intensity vs. chemical constitution of EOs from vegetative parts of *Ocimum basilicum* L. (Lamiaceae), it was evidenced that, when the 1,8-cineole content decreased, the amount of linalool increased significantly [137]. This fact can also be explained by the increase in temperature that causes greater volatilization of linalool, which can stimulate the biosynthesis of oxygenated cyclic monoterpenes such as camphor and 1,8-cineole [62].

Rainfall intensity and relative air humidity also have a strong influence, directly proportional, on the chemical constitution of some monoterpenes and sesquiterpenes in the EOs of *P. mollicomum*. There are not many studies that list convergences between rainfall intensity and volatile constituents. However, some analyses have related this climatic variable with the production of sesquiterpenes in the EOs of several plant species [138,139,140,141,142]. These results support our findings that higher rainfall intensities may be associated with the production of some terpenes, such as α-terpinene.

Our data also propose to discuss the role of the environmental variable wind speed in the biosynthesis and consequent emission of volatile constituents in *P. mollicomum*. The results showed that the content of some volatiles, such as *Z-β*-ocimene, eupatorium-chromene, benzyl benzoate, *α*-terpinene, tolualdehyde and *δ*-cadinene, is positively influenced when the wind speed increases. There is some scarcity in the literature on the referred correlation, mainly, of the direct action of the wind on the content of the chemical constituents of the EOs of vegetative structures. However, a recent study carried out with two important tree species, *Betula pendula* Roth and *Betula pubescens* Ehrh. (Betulaceae), suggested that emission rates of monoterpenes and sesquiterpenes increased when the wind speed was measured, at relatively high rates in *Betula pendula* Roth [143]. More tests evaluating the circadian cycle, along with continuous measurements of microclimatic variables, may aid in the more efficient interpretation of these possible correlations. However, our first findings already indicate that abiotic factors can influence the modulation and biosynthesis of terpenes, which may influence the attraction and repellency of insects.

Our additional objective was to evaluate the following question: do microclimatic environmental variables jointly interfere in the activity of visits by potential pollinators to inflorescences? Our results related to the Spearman’s correlation between microclimatic abiotic variables vs. frequency of visits by potential pollinators, showed a moderate positive correlation between air temperature, inflorescence temperature, leaf temperature, relative air humidity and rainfall index, by insects 2 (Hymenoptera—*Tetragonisca angustula* Latreille., 1811), 3 (Diptera—*Syrphidae* sp. 2), 5 (Hymenoptera—*Halictidae* sp. 2) and 6 (Hymenoptera—*Colletidae* sp.), in addition to a weak positive correlation for light intensity, and a frequency of visits by insects 4 (Hymenoptera—*Halictidae* sp. 1) and 5 (Hymenoptera—*Halictidae* sp. 2). Rain and humidity can bring disadvantages to insect flight, meaning a decrease in visitation density [76,77,78]. These results also differ from previous studies, as they indicated that insect three (3), a dipteran, showed greater visitation activity in relatively warm periods. These studies, in addition to addressing the inefficiency of flies as pollinators, argue that these insects showed a higher frequency of visitation during seasons when ambient temperature and radiation were lower [144]. The frequency of visits by most pollinators, particularly bees, depends on specific values of microclimatic variables to carry out their activities [145,146]. There are few studies that purport to examine how temperatures influence Hymenoptera survival rates, however, in the 1990s, some studies found that the pollination process performed better when bees were exposed to relatively higher temperatures [147]. Some of these insects, especially those of the genus *Apis*, are active and extremely effective pollinators. Foraging typically begins in the morning when the lowest values of temperature, light intensity, and radiation are measured to be between 15.5 and 18.5 °C, 600 to 1700 lx, and 9 to 20 mW/cm^2^, respectively [148,149].

In addition to temperature, radiation and light intensity, another factor that influences the activities of insects that visit flowers is the altitude [76,77,150,151,152]. Some bees exhibit specialized patterns of activity when flying, which makes them more adept at controlling their body temperature. Compared to others who do not have or use the same routines as them, they are more robust to temperature variations [153]. These findings, in addition to corroborating our results, demonstrate that some insects can obtain advantages in the competition for food in relation to others, as they have thermoregulatory physiological strategies, which contribute to the promotion of food collection in the inflorescences even in extreme conditions of temperature. Our data demonstrate that insect two (2), bee *T. angustula* (jataí), had much higher visitation rates than other potential pollinators, which is similar toother results [154,155,156].

In the times under observation, *T. angustula* was the main floral visitor. This bee is considered a sociable forager, which uses a variety of chemical communication strategies with its partners, signaling them where to find rich food sources [79,80]. Given the results of positive correlations between volatiles vs. insect visits and data from Appendix A, we can infer that phenological event patterns are not so expressive for the density of visits by these bees in search of food reward. This observed episode may be due to the insect–insect signaling behavior among individuals of the *T. angustula* society [79,80].

Concluding our analyses and correlations, we propose to relate whether microclimate meteorological variables influence the patterns of phenological events. The data showed a strong negative correlation between leaf fall and temperature; a strong positive correlation between rainfall and leaf sprouting; in addition to a strong negative correlation between light intensity and budding. Some published studies have shown that sudden drops in temperature can cause lesions and photoinhibition in the leaves of plant species, directly and indirectly, increasing the leaf abscission process, which corroborates our findings [43,157,158]. In addition, it is known that exposing plants to high light intensity, and prolonged periods of water stress, can cause harmful changes to the flowering cycle of some species [159,160]. However, there are few studies in *Piper* related to these variables.

## 4. Materials and Methods

### 4.1. Area of Study

This research was carried out weekly, from September 2020 to January 2021, in the Tijuca National Park (TNP) (43°14′29.64″ W/22°58′9.80″ S), at an elevation that varies between 127 and 68 m between the studied specimens. The TNP is located in the South Zone of the City of Rio de Janeiro and has a tropical monsoon climate, according to the Köppen–Geiger classification [161]. Six adult specimens, with an average height of 1.65 m in an open area, were separated as samples from a population of *Piper mollicomum* Kunth (PM) from this region. Licenses for the investigations were granted by SISBIO (number 57296-1; authentication code 47749568). The surroundings of the experimental plot consisted of native vegetation. Fertile specimens were previously identified by taxonomist Dr. George Azevedo de Queiroz at Botanical Garden of Rio de Janeiro (JBRJ), and the samples were deposited in the Herbarium of the University of the State of Rio de Janeiro (HRJ/UERJ) (Appendix A).

### 4.2. Climate Data

Data on meteorological microclimate variables for the days of weekly collections were obtained with the aid of manual meters: *Digital Windmeter Anemometer* (SIN2919025384—Brazil, Rio de Janeiro)—wind speed, temperature and relative air humidity; measured in the same period of observation of the frequency of visits by potential pollinators to mature inflorescences of *P. mollicomum*; *Luxmeter* (INSTRUTEMP. 1,712,268—Brazil, Rio de Janeiro)—luminosity; performed once at each time of observation of the frequency of visits; and *Infrared Laser Thermometer* (EXBOM—TDI 330 – Brazil, Rio de Janeiro)—for measuring the surface temperatures of leaves and reproductive organs; also collected once at each visit time. Therefore, the environmental variables measured were: (a) air temperature (°C), (b) inflorescence and infructescence temperature (°C), (c) leaf temperature (°C), (d) relative humidity (%), (e) wind speed (m/s), (f) rain (mm) and (g) light intensity (kJ/m^2^).

The climatological data referring to the pluviometric indices were obtained from the Meteorological Station of Forte de Copacabana in the City of Rio de Janeiro, at the Brazilian Institute of Meteorology (INMET), in the weeks in which the observations took place. The sets of climatological variables were tabulated for linear correlation with other data: (a) frequency of insect visits to mature inflorescences; (b) chemical constituents identified in the EOs from leaves and from reproductive organs (five stages) and (c) phenological events.

### 4.3. Reproductive and Vegetative Phenological Study

The qualification and quantification of *P. mollicomum* phenology were carried out in the six individuals in the aforementioned period. The phenological events evaluated were:(a) leaf sprouting (emissions from vegetative buds); (b) leaf fall (reduction of leaves on stems or branches); (c) immature inflorescences or buds (emission of green or yellow inflorescences, without the presence of pollen, and with closed bracts); (d) mature inflorescences (anthesis flowers with the presence of pollen) and (e) immature infructescences (process ranging from the oxidation of mature inflorescences to the presence of infructescences with hardened pericarps). Due to the small size of the flowers and fruits, a manual lens (60X magnification) was used to observe the inflorescences and infructescences in the field, in addition to determining which flowers were in anthesis [162].

Fournier Intensity (FI) methodology was used to quantify the phenophases mentioned above [163]. The FI consists of the formation of a scale that groups the phenological patterns analyzed in the plant according to the following criteria: 0—absence of the investigated phenophase; 1—presence of the investigated phenophase in the proportion of 1 to 25%; 2—presence of the investigated phenophase in the proportion of 26 to 50%; 3—presence of the investigated phenophase in the proportion of 51 to 75% and 4—presence of the investigated phenophase in the proportion of 76 to 100%.

### 4.4. Frequency of Visits by Potential Pollinators

Observations of the frequency of visits by potential pollinators of *P. mollicomum* were performed weekly, in the period and in locus described above, from 8:00 am to 5:00 p.m. (with 30 min of observation, and 30 min of resting, totaling 128 h). During this period, plant specimens contained significant amounts of mature inflorescences [164,165]. The frequency of visits was performed by counting the number of times that the insects visited the inflorescences in the period [164,165]. The floral visitors were collected, for posterior identification, with the aid of an entomological net, with approximately 35 cm in diameter of the basket, 80 cm in depth (12 mm of mesh), and a reach of 3 m. The insects were promptly anesthetized in flasks containing cotton slightly moistened with a 70% (v/v) hydroalcoholic solution and then, to preserve the samples, they were placed in individual packages containing the same solution [166]. Specimens were total or partially identified using a reference collection from the Rio de Janeiro Botanical Garden. The main visitors described were: Insect 1 (Diptera—*Syrphidae* sp. 1); Insect 2 (Hymenoptera—*Tetragonisca angustula* Latreille. 1811); Insect 3 (Diptera—*Syrphidae* sp. 2); Insect 4 (Hymenoptera—*Halictidae* sp. 1); Insect 5 (Hymenoptera—*Halictidae* sp. 2) and Insect 6 (Hymenoptera—*Colletidae* sp.).

### 4.5. Sample Colletion to Essential Oil Extraction

Leaves (150 g) and five different stages of the reproductive organ of *P. mollicomum* (5 g to 40 g) were collected monthly in the reproductive period, in the morning, between 9:00 and 10:00 a.m., for analysis involving ontogenic vs. constituents of the EOs. These materials were manually reduced with scissors and were submitted to the hydrodistillation method in a modified Clevenger-type apparatus. The reduced material was transferred to a 2 L round bottom flask, to which 700 mL of distilled water was added. The hydrodistillation process was carried out for 2 h. Upon completion, the pure EOs were separated from the aqueous phase, subjected to drying with anhydrous sodium sulfate (Sigma-Aldrich, Brazil) and stored in a closed amber vial, in a freezer at −20 °C, until the time of analysis [2,3,55]. Yields were calculated as the volume ratio in milliliters of EOs divided by the weight in grams of fresh plant material used in the extraction, subsequently multiplied by 100 (percentage content. *v*/*w*) [3,55]. The five stages of the reproductive organs of *P. mollicomum* (Figure 6) were described by taxonomists Dr. Elsie Franklin Guimarães and Dr. George Azevedo de Queiroz, both from the Research Institute of the Rio de Janeiro Botanical Garden. The different stages were characterized as:

**Stage 1:** Inflorescence in spike; presenting very immature development since its floral elements are protected by the bracts still in formation, arranged in inclined bands, with green color from the peduncle to the inflorescence.

**Stage 2:** Inflorescence in spike; presenting an immature stage of development since its floral elements are still protected by bracts in formation, arranged in bands inclined to horizontal. The length of the peduncle is slightly longer and slightly changes its hue to yellowish-cream.

**Stage 3:** Inflorescence in spike; presenting a more advanced stage of development from bottom to top since its floral elements are already more defined, where the triangular bracts (arranged in horizontal bands) and open anthers with the release of pollen grains are morphologically visualized. The length of the peduncle is longer and the inflorescence changes its hue to intense yellow.

**Stage 4:** Inflorescence in spike; presenting a more advanced stage of development from bottom to top since its floral elements are already more defined, where the triangular bracts (arranged in horizontal bands) and open anthers with the release of pollen grains are morphologically visualized. The peduncle length is continuously longer, still green, while the inflorescence changes its tonality to yellow/brown interspersed bands, due to the presence of mature anthers.

**Stage 5:** Inflorescence in spike; presenting initial stage of fruit in all its extension since one of the floral elements, the ovary, presents itself modified in its formation. At this stage, the triangular floral bracts are morphologically visualized, anthers already senescent, which may or may not keep their filaments adherent to the fruit, which is in development; arranged in horizontal bands, maintaining the length of the peduncle that still remains green, while the inflorescence changes its tonality to intense green.

### 4.6. Essential Oil Analysis

The EOs obtained by hydrodistillation were solubilized in dichloromethane (HPLC grade, Tedia, Brazil – Rio de Janeiro) to obtain a final concentration of 1000 ppm. Subsequently, these diluted EOs were submitted to analysis by Gas Chromatography coupled to Mass Spectrometry (GC/MS) in an HP Agilent CG 6890—EM 5973N Equipment – Rio de Janeiro, to identify the constituents by their respective mass spectra. To determine the quantitative parameters of the EOs and calculation of the Retention Index (RI), the analyses were performed by Gas Chromatography Coupled to the Flame Ionization Detector (GC/FID), in an HP-Agilent 6890–Rio de Janeiro equipment. The GC/MS analyses were: HP-5MS capillary analytical column (HP Agilent CG 6890—EM 5973N Equipment–Brazil, Rio de Janeiro (30 m × 0.25 mm i.d. × 0.25 µm film thickness) and temperature ramp from 60 °C to 240 °C, with an increment of 3 °C/min, helium (~99.9999%) as carrier gas, at a constant flow rate of 1.0 mL/min; scanning range of masses (*m*/*z*) at 40—600 atomic mass units (u), with impact energy of 70 eV, operating in positive mode. A 1 μL sample of the EOs solution was injected, in splitless flow mode, and with the injector temperature at 270 °C [3,167]. GC/FID analyzes were performed on an HP-5MS capillary analytical column (30 m × 0.25 mm i.d. × 0.25 μm film thickness), with the same temperature program used for GC/MS analyses, but with hydrogen as a carrier gas at a constant flow rate of 1.0 mL/min. The EOs solution by GC/FID was injected under the same conditions as GC/MS. Retention times (Rt) were measured in minutes without correction, and the relative percentage of each compound was determined by the signal area [2,3,55,167,168]. The RIs were calculated from the results of the analysis of a homologous series of saturated aliphatic hydrocarbons (C_8_-C_28_, Sigma-Aldrich, Brazil), performed in the same column and conditions used for the analysis by GC/FID. The identification of constituents was made by comparing the mass spectra obtained from data from the equipment’s library (NIST and Wiley https://webbook.nist.gov/chemistry/cas-ser/ (accessed on 1 January 2021)), and records in the literature [169]. All analyses were performed in triplicate and results are presented as average standard deviation (Appendix A).

### 4.7. Statistical Analysis

Statistical treatments were performed using the Statistica software version 10 (StartSoft Inc., Tulsa, OK, USA). Spearman’s correlation test was processed to assess the relationships between abiotic factors, as well as volatile constituents vs. ontogeny vs. frequency of visitation of potential pollinators. The correlation coefficient of this inspection can be positive (directly proportional) or negative (inversely proportional), assimilating quantitative values that suggest that the relationship between two or more variables is strong, moderate or weak: 0.00 to 0.19 (very weak); 0.20 to 0.39 (weak); 0.40 to 0.69 (moderate); 0.70 to 0.89 (strong) and 0.90 to 1.00 (very strong) [56]. Canonical correspondence analyses were used to evaluate possible correlations between EO constituents from leaves and mature inflorescences vs. frequency of visits by potential pollinators. The graphs of the results were correlated by approximation, that is, the closer the compounds are to the insects, the greater the correlation between the variables. The chemometric analyses, including the principal components analysis (PCA), were performed in the R software (version number: 4.1.0; created by Ross Ihaka e Robert Gentleman; Department of Statistics at the University of Auckland, New Zealand) and designed to evaluate the variation between the EOs of different organs in different collection periods [170].

## 5. Conclusions

All correlations between volatiles vs. potential pollinators visits vs. ontogeny vs. microclimate vs. phenology allow us to convey important data regarding the complex information network in *Piper mollicomum*, a medicinal and ritualistic plant of fundamental value for the regeneration of deforested areas, such as Atlantic Forest. This is the first scientific work that studies the reproductive period of this *Piper* species. As a result, the information gathered is fundamentally important for comprehending the mechanisms of chemical–ecological plant–insect interactions in Piperaceae, a basal angiosperm.

## Figures and Tables

**Figure 1 plants-11-03535-f001:**
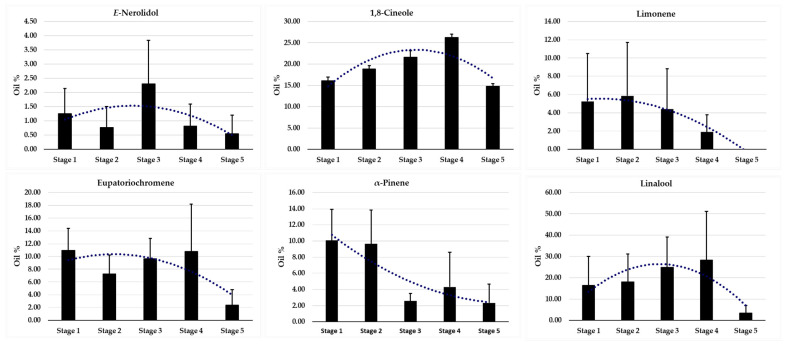
Percentage content (mean standard deviation) of the major compounds vs. stages of development of the reproductive organ of *Piper mollicomum* Kunth, in the months of September 2020 to January 2021. For the construction of this graphics, we selected only the constituents that registered percentage contents above 5% at least once in the EOs in the periods and stages of the analyzed reproductive organs.

**Figure 2 plants-11-03535-f002:**
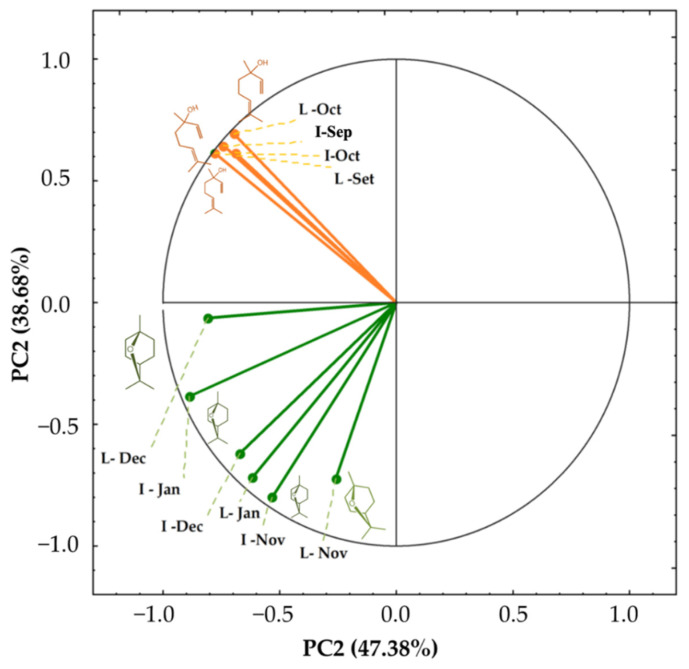
Ordering diagram produced by Principal Component Analysis (PCA) relating the chemical constitution of essential oils from leaves and inflorescences of *Piper mollicomum* Kunth in the months under study (September 2020 to January 2021). Sep–September; Oct–October; Nov–November; Dec–December; Jan–Jan. L–Leaves; I–Inflorescences.

**Figure 3 plants-11-03535-f003:**
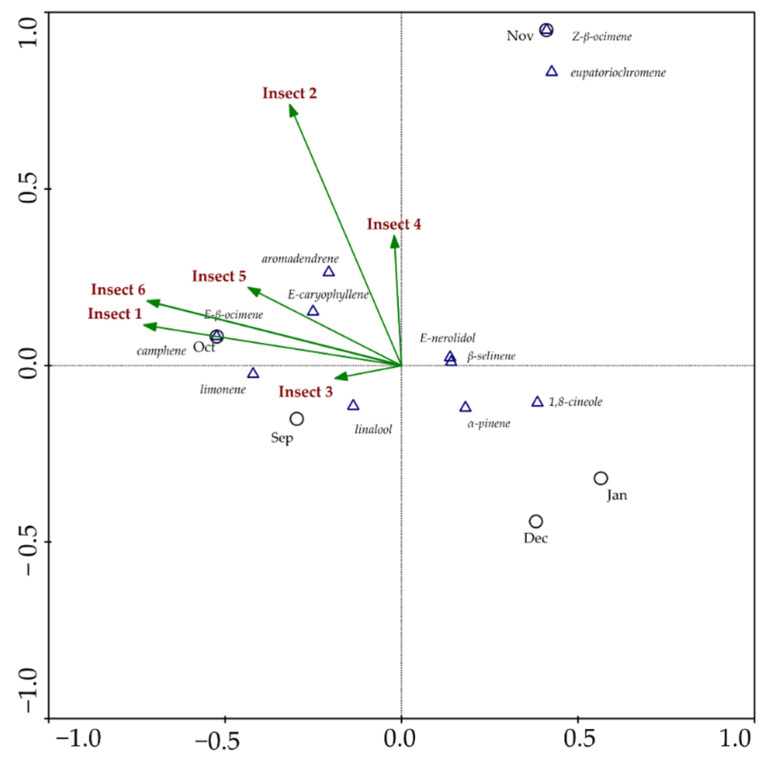
Canonical Correspondence Analysis (CCA) that correlates the chemical composition of foliar essential oils and the frequency of visits of potential pollinators of *Piper mollicomum* Kunth from September 2020 to January 2021. Sep–September; Oct–October; Nov–November; Dec–December; Jan–January; Insect 1 (Diptera—*Syrphidae* sp. 1); Insect 2 (Hymenoptera—*Tetragonisca angustula* Latreille. 1811); Insect 3 (Diptera—*Syrphidae* sp. 2); Insect 4 (Hymenoptera—*Halictidae* sp. 1); Insect 5 (Hymenoptera—*Halictidae* sp. 2); Insect 6 (Hymenoptera—*Colletidae* sp.).

**Figure 4 plants-11-03535-f004:**
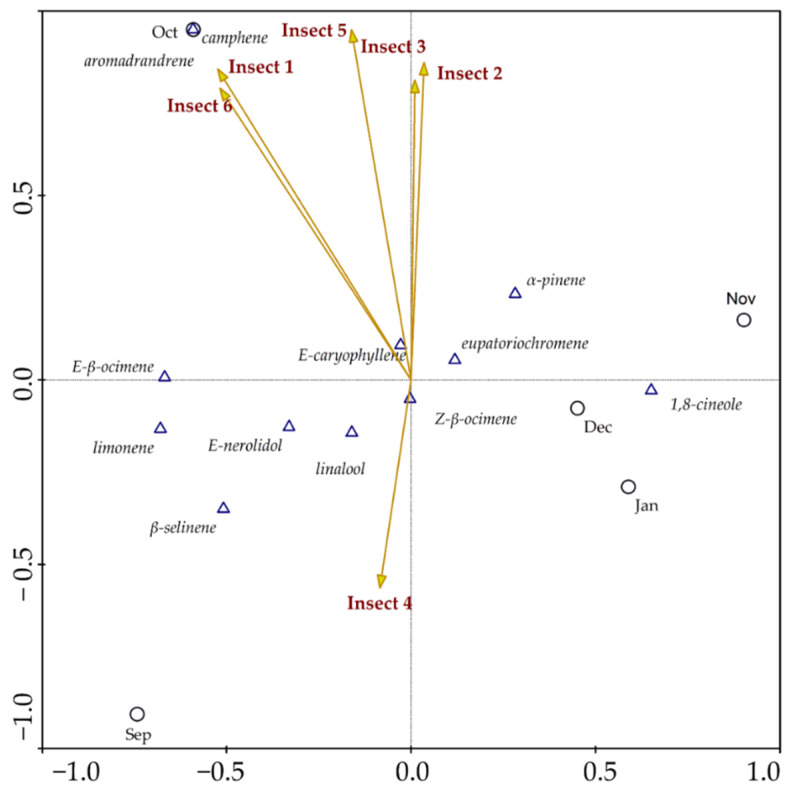
Canonical Correspondence Analysis (CCA) that correlates the chemical composition of essential oils from mature inflorescences and the frequency of visits by potential pollinators of *Piper mollicomum* Kunth from September 2020 to January 2021. Sep–September; Oct–October; Nov–November; Dec–December; Jan–January; Insect 1 (Diptera—*Syrphidae* sp. 1); Insect 2 (Hymenoptera—*Tetragonisca angustula* Latreille. 1811); Insect 3 (Diptera—*Syrphidae* sp. 2); Insect 4 (Hymenoptera—*Halictidae* sp. 1); Insect 5 (Hymenoptera—*Halictidae* sp. 2); Insect 6 (Hymenoptera—*Colletidae* sp.).

**Figure 5 plants-11-03535-f005:**
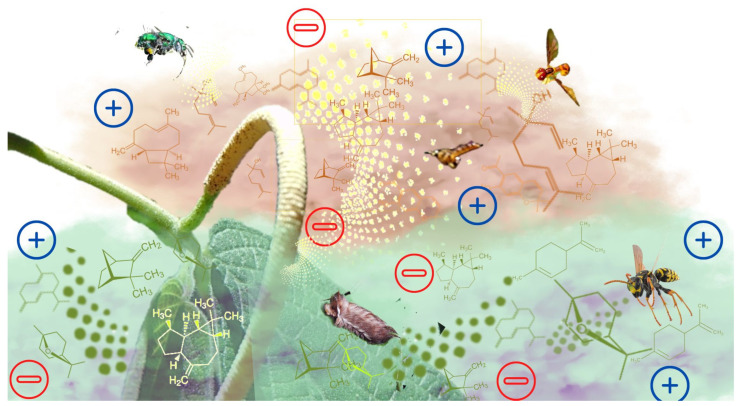
Illustration of possible harmonic and disharmonic interactions provided by the attraction and repulsion of volatile constituents contained in essential oils: an illustration of the volatile fog. These compounds interact in a blend or volatile mixture, a phenomenon that was already described.

**Figure 6 plants-11-03535-f006:**
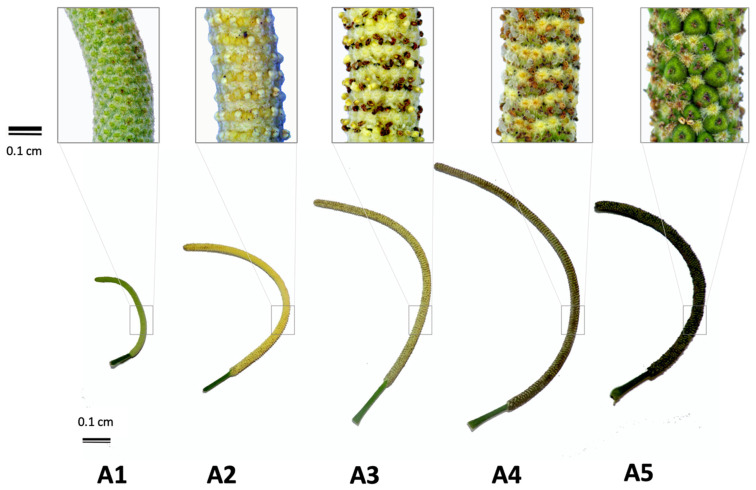
Stages of floral development of *Piper mollicomum* Kunth from the beginning of its formation (A1) to maturity (A5). A1—Inflorescence stage 1; A2—Inflorescence stage 2; A3—Inflorescence stage 3; A4—Inflorescence stage 4; A5—Inflorescence stage 5. For stage A1 to A5 description.

## Data Availability

Not applicable.

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
