# Peer review of "Volatile Chemical Variation of Essential Oils and Their Correlation with Insects, Phenology, Ontogeny and Microclimate: *Piper mollicomum* Kunth, a Case of Study"

_plants, 2022, doi:10.3390/plants11243535_

Round 1
Reviewer 1 Report
Comments for manuscript plants-2000905
General comments:
The manuscript “Volatile chemical variation and their correlation with insects, phenology, ontogeny and microclimate: Piper mollocomum Kunth, a case study”,
presents an interesting approach on studying and try to understand the complex interaction of a host plant, its pollinating insects and the environment, all based on changes in odour profiles of the plant. A creative way of analyzing the data has been used. It is a very nice piece of work and I would like to see it published!
In general, the manuscript is well-written and the data-analyses and figures/tables are appropriate and inventive. Nevertheless, the manuscript must be rewritten carefully regarding typing errors/slips of the pen (?). Also, terms regarding ecology of insect-plant interactions / tri-trophic interactions and odor-mediated host plant recognition should be used correctly and consistently. Some additional explanations here would help the reader (see minor comments). Parts of results and discussion are repetitive and should be trimmed. Throughout the manuscript the statements on the results and/or conclusions must be formulated more carefully. The results indicate possible mechanisms – yes –, but they did not show or even proof them!
The discussion is too long and quite difficult to follow for the reader. There is a lot of information given, but please cut to the basics. It is too much information to follow the central theme. The discussion should be rewritten completely. Present only information needed to discuss the results.
Minor comments:
p. 1, line19: Kunth, not Kunt
p.1, line 25: attraction, not atraction
p.1, line 38: cyophilous species?
p.1, line 38-44: please add the climate zone of this species here in the Introduction, possibly also details on preferred climate
p.2, line 58: C. castanea in italic
p.2, line 62-63: …, a current study has focused on the understanding…OR …, current studies have focused on the understanding…;
not…, current studies has focused on the understanding…
p.2, line 70: please use IUPAC name instead of “chromenes”
p.2.line 81: voracious herbivory caterpillars instead of “voracious predatory caterpillars” (Predators are animals that eat other animals, called prey)
p.2, line 84: …avoid herbivory… instead of “…avoid predation…” (see comment above) !!! throughout the whole manuscript: use the correct term herbivory/predation etc. !!!
p.2, line 86-87: rephrase and clarify “…emit EOs attractive to opportunistic wasps that feed on their phytophagous opponents, promoting…” Please state clearly what kind of wasps you are talking about, a parasitic wasp? a parasitoid? a predator?
p.2, line 88-89: same in this sentence, rephrase and clarify if the wasps is a parasitoid or parasitic wasp “This mechanism infers that some constituents may benefit the plant by attracting opportunistic insects that prey on their natural enemies [24].”
p. 3, line 101: rewrite …, aromatic volatiles are constituents used since…., not “…, volatiles are aromatic constituents used since antiquity, being included…” (not all volatiles are aromatic)
p. 3, line: 106: rewrite “predators of plants” (see comments above: Predators are animals that eat other animals, called prey. Animals that eat on plants are called herbivores)
p. 3, line 107-108: rewrite: “In addition, they can even prevent the oviposition of natural plant enemies, and attract predatory insects of these animals [52,68,69]”.
What is a “natural plant enemy”?!
Better: “In addition, they can even prevent the oviposition of herbivore insects/animals, and attract predators of these insects/animals [52,68,69]”
p.3, line 109-111: rephrase these two sentences. The relationship between the prey and predator, a herbivory insect and a plant is not “negative”. “harmonious association between these to groups” is also a strange wording. There are terms for these different kinds of allelochemicals mediating interactions between different species: kairomones, allomones and synomones. It helps the manuscript to use them here.
p.3, 116-131: please explain shortly WHY your work is “of great chemical-ecological importance”. For what can you use the results? Which mechanisms possible based on your results?...
Results
Some parts stated in the result section belong in the discussion. Please rewrite carefully!
p.3, line 133-135: Difficult to understand what you mean. Please clarify what you mean with this first sentence of the Results.
Add “all” : “…of all five stages…”?
p.5-7, table 1 and 2: add CAS numbers for proper ID of compounds; please add in table/footnote – and in Material and Methods - how you calculated “relative percentage %”
p.8, line 182-184: please explain shortly how you select the 6 “main compounds” presented in Figure 2. Your Spearman’s correlation was significant for 3 compounds only, and myrcene is not part of Figure 2? Please clarify here what you did.
p.9, 210-215 + p.11 Figure 4: Please explain how you can see what you stated in the paragraph p.9, 210-215 in Figure 4. It is difficult to follow you here.
p.10, line 234-242 + p.12 Figure 6: The order of your 6 insects is confusion (Why not insect 1 the most common visitor and insect 6 the one with less visits?)
p.14, line 299-302 + p. 16 Figure 9: There are some theories on the mechanisms behind odor-mediated host plant recognition and your “volatile fog”. It is a nice term, but it must be stated that the phenomenon is known already. It would help the manuscript to give an overview about the theories of mechanisms and proofs for them here or rather in the discussion. Perhaps move figure 9 + overview on the theories of mechanisms completely in the discussion?
Plant odors are complex and dynamic blends that consist of both, chemicals common to many plant species and more species-specific compounds. A more recent and strongly supported hypothesis proved ratio-specific odor recognition, using particular ratios of volatiles that are common in several plant species. Spatiotemporal coincidence detection of insects during host location is an additional theory. Spatiotemporal coincidence detection is based on the ability of insects to identify host volatile blends within complex and dynamic background odors emitted by surrounding vegetation. Volatiles perceived by an insect in the context of the correct blend, i.e., in spatiotemporal coincidence, are identified as host signals irrespective of the background odor, whereas the same volatiles sensed individually outside the context of the correct blend are perceived as non-host cues. And there is a newer theory including volatile cues from the habitat…
You might find more details e.g. here:
Bruce, T. J. A. Interplay between insects and plants: dynamic and complex interactions that have coevolved over millions of years but act in milliseconds. J. Exp. Bot. 2015, 66, 455−465.
Bruce, T. J. A.; Pickett, J. A. Perception of plant volatile, blends by herbivorous insects − finding the right mix. Phytochemistry 2011, 72, 1605−1611.
Bruce, T. J. A.; Wadhams, L. J.; Woodcock, C. M. Insect host location: A volatile situation. Trends Plant Sci. 2005, 10, 269−274.
Randlkofer, B.; Obermaier, E.; Hilker, M.; Meiners, T. Vegetation complexity – the influence of plant species diversity and plant structures on plant chemical complexity and arthropods. Basic Appl. Ecol. 2010, 11, 383−395.
Schroeder, R.; Hilker, M. The relevance of background odor in resource location by insect: A behavioural approach. Bioscience 2008, 58, 308−316.
Visser, J. H. Host odour perception in phytophagous insects. Annu. Rev. Entomol. 1986, 31, 121−144.
Webster, B., Cardé, R. T. Use of habitat odour by host-seeking insects. Biol. Rev. 2017, 92, 1241–1249.
Webster, B.; Gezan, S.; Bruce, T.; Hardie, J.; Pickett, J. Between plant and diurnal variation in quantities and ratios of volatile compounds emitted by Vicia faba plants. Phytochemistry 2010, 71, 81–89.
p.19, line 372-374: Define “Pattern of phenological events” here, to clarify your results. There are different types of visual cues e.g. colour, shape etc. different insects use different types of visual cues… write more about that. Perhaps your results are not so “surprising” after that. Please add more details on types of visual cues also in Material and Methods. How did you define them? Why did you use pattern? How did you “measure” the pattern?...
p.19, line 377-379: “Given the results of positive correlations between volatiles vs. insect visits and data from Table 6, we can infer that visual stimuli are not so expressive for the density of visits by these bees in search of food reward.»
Be careful with this statement! See comment above. You did not investigate visual cues properly, thus you can not state that they have a minor importance. Rephrase this part.
The discussion is too long and difficult to follow for the reader. There is a lot of good information given, but please cut to the basics. The discussion should be rewritten completely. Present only information needed to discuss the results. State the facts only, delete details not directly related to your study. E.g., a lot of examples on studies on odour-based pest insect-plant interactions. Focus on odour-based pollinator-plant interactions (1) and odour-based natural enemy – plant interactions (2), as you found these insect groups in your study. Another example (p. 23-24): You describe Linalool and its role in plant-insect interaction on approx. 1 page. Facts, interesting for your study, can be described in 2-3 sentences. Please reduce text.
Reviewer 2 Report
1- In page 1 , line 2,3 : Volatile chemical variation of essential oils and their correlation with insects, phenology, ontogeny and microclimate: Piper mollicomum Kunth, a case of study instead of Volatile chemical variation and their correlation with insects, 2 phenology, ontogeny and microclimate: Piper mollicomum 3 Kunth, a case of study
2- In page no. 1 line 21 Gas Chromatography / Mass Spectrometry (GC/MS) instead of Gas Chromatography (GC) coupled to Mass Spectrometry (MS)
3- In page 2 line 37-131: Introduction is so long and should be summarized
4- Page 3 lines 135-138 : The EOs throughout the five stages of development of reproductive organs showed variation in their percentage of volatile chemical compositions (0.01% to 1.47%; v/w). Stages 1, 2, 3 and 4 afforded the lowest percentages (0.01% to 1.12%; v/w) and stage 5 the highest (1.2% to 1.47%; v/w) (Tables 1 and 2) instead of The EOs throughout the five stages of development of reproduc- 135 tive organs showed variation in their yields (0.01% to 1.47%; v/w). Stages 1, 2, 3 and 4 136 afforded the lowest percentages (0.01% to 1.12%; v/w) and stage 5 the highest (1.2% to 137 1.47%; v/w) (Tables 1 and 2)
5- In page 5,6 table one and table 2: one column contain the type of compound (Monotrepene, Oxygenated monoterpene, sesquterpene , oxygenated sesquterpene and non oxygenated sesquterpene) should be add because the author added that in pages 3 line 138-141 [The volatile constituents were different in most of the inves- 138 tigated months, demonstrating a rich and diversified fraction in non-oxygenated sesquit- 139 erpenes in the leaves and oxygenated monoterpenes in the five stages of the reproductive 140 organ. As example ] to easily understand.
6- Page 3 line 149: what is the meaning GC-MS
7- Page 31 line 954 the author write all analyzes of essential oils were performed in triplicate and results are presented as average but standard deviations not appear in table one and two as example explain why?
Reviewer 3 Report
Dear colleagues,
The topic of the paper presented by Machado et al. may help to understand the interaction between plants and insects. The results are based on insects sampling and collection of volatiles.
-The paper is very very long and descriptive, making it difficult to follow and flow when reading. I feel that I read a thesis and not a research paper. The message of introduction as well as the interpretations are not clear and not easy to know what they need. The authors move regularly from specific points to more general ones.
The authors presented and discussed their results in “Result section” with references. Then they have a separated discussion
Thus, introduction, results and discussion need complete MAJOR revisions. Besides, it is important that authors should avoid the overinterpretation of their data in discussion, especially regarding the attraction of insets towards the plants. This needs further behavioural experiments.
-The authors need to reconsider the statistical analysis. They used spearman's correlation test assess the relationships between abiotic factors, as well as volatile constituents vs. ontogeny vs. frequency of visitation of potential pollinators. I propose to use linear regression analysis to figure out the impact of abiotic factors on volatile and pollinator. So, you set out various models according to the weather condition. I mean, you have 8 abiotic factors, then you would have around 64 models. The best model would be defined according to the lowest Akaike information.
-I also encourage to seek a help for English editing.
Few minor comments (there are much more):
L63: delete (plant, including)
L86: Replace “EOs” by “the Essential oils”. Full name in the first mention in introduction.
L107-108: replace “In addition, they can even prevent the oviposition of natural plant ene-107 mies, and attract predatory insects of these animals” by “In addition, they can even prevent the oviposition of insects and mites, and attract their natural enemies
L116: replace “Ramos and collaborators” by “Ramos et al” Idem for all next
L116-131: completely lost in this paragraph. Rephrase it.
L134: from December 2020. However, in Table 1, you mentioned September
L880: How did you collect the insects.
L967-969: More information is needed regarding the analysis
Figure 1: move it to material and methods “Sample collection to essential oil extraction”
Delete Figure 6. The images of insects are not clear. It is enough to mention them in results.
Delete Figure 9. In my opinion, this image would be as graphical abstract or if the paper were a review.
Round 2
Reviewer 2 Report
None
Reviewer 3 Report
-Although, the authors have improved their paper, it’s still not ready for publication. In the first version, the paper contained 39 pages and now it contained 38 pages. They moved one figure to supplementary materials. Again, this is not a thesis, so the authors must shorten and polish the paper (particularly discussion), so it makes so much more succinct and easier to read.
-The authors acknowledged in discussion that some compounds emitted from Piper mollicomum are attractive to natural enemies because of infestation. However, I did not see any survey of record for insect herbivores in this study (correct me if am wrong).
-The authors replied to my previous comments regarding the Spearman's correlation by providing few papers (of one of co-authors) that used the same approach of analysis. The Spearman's correlation is a correlation between two variables. In the current study, you used many factors so it’s important to interact all factors together, the reason why I suggested to use linear regression analysis. Kindly see again my previous comment below:
(The authors need to reconsider the statistical analysis. They used spearman's correlation test assess the relationships between abiotic factors, as well as volatile constituents vs. ontogeny vs. frequency of visitation of potential pollinators. I propose to use linear regression analysis to figure out the impact of abiotic factors on volatile and pollinator. So, you set out various models according to the weather condition. I mean, you have 8 abiotic factors, then you would have around 64 models. The best model would be defined according to the lowest Akaike information).
-L95-97: delete
-L111-114: Rephrase it (Furthermore, in a sequence of studies by our group, which evaluated the chemical constitution of EOs from different organs of P. mollicomum, showed a high percentage of linalool, eupatorio chromene and E-nerolidol, in addition to other major constituents, in the reproductive structures [ 2,3]).
-L114-115: (However, in these studies, macroclimatic data were used, and the different phenophases of the reproductive organ were not evaluated). I did not understand here: does macroclimatic data was or not in these studies?
-L116: delete “motivated by the initial results”
-L118-122: you can start this paragraph in discussion (that you need to shorten the 6 pages).
-L333-336: delete
-L346-347: delete “According to 346 our results”
-The abbreviations in figure 4 are written in Portuguese. Check them and idem for other figures?
